# The associations of alcoholic liver disease and nonalcoholic fatty liver disease with bone mineral density and the mediation of serum 25-Hydroxyvitamin D: A bidirectional and two-step Mendelian randomization

Qinyao Huang[1☯], Jianglong Guo[2☯], Hongjun Zhao[1], Yi Zheng[3], Yuying Zhang[4]*

1 The Sixth Affiliated Hospital of Guangzhou Medical University (Qingyua People's Hospital), Qingyuan, China, 2 Department of Medical Imaging, The Second Clinical School of Guangzhou Medical University, Guangzhou, China, 3 Department of General Surgery, The First Affiliated Hospital of Harbin Medical University, Harbin, China, 4 Central Laboratory, Shenzhen Longhua Maternity and Child Healthcare Hospital, Shenzhen, China

☯ These authors contributed equally to this work.
* yuying_zhang@163.com

**Data Availability Statement:** The datasets analyzed during the current study are available in

## Abstract

### Background

Reduced bone mineral density (BMD) and osteoporosis are common in chronic liver diseases. However, the causal effect of alcoholic liver disease (ALD) and non-alcoholic fatty liver disease (NAFLD) on BMD remains uncertain.

### Objectives

This study uses a two-sample Mendelian randomization (MR) design to evaluate the genetically predicted effect of ALD and NAFLD on BMDs using summary data from publically available genome-wide association studies (GWASs).

### Methods

The GWAS summary statistics of ALD (1416 cases and 213,592 controls) and NAFLD (894 cases and 217,898 controls) were obtained from the FinnGen consortium. BMDs of four sites (total body, n = 56,284; femoral neck, n = 32,735; lumbar spine, n = 28,498; forearm, n = 8143) were from the GEnetic Factors for OSteoporosis Consortium. Data for alcohol consumption (n = 112,117) and smoking (n = 33,299) and serum 25-Hydroxyvitamin D (25-OHD) level (n = 417,580) were from UK-biobank. We first performed univariate MR analysis with the Inverse Variance Weighted (IVW) method as the primary analysis to investigate the genetically predicted effect of ALD or NAFLD on BMD. Then, multivariate MR and mediation analysis were performed to identify whether the effect was mediated by alcohol consumption, smoking, or serum 25-OHD level.

the FinnGen repository (http://r5.finngen.fi/) (finn-b-K11_ALCOLIV and finn-b-NAFLD), UK Biobank (http://www.nealelab.is/uk-biobank) (ieu-a-1283, ukb-b-469,and ebi-a-GCST90000617) and GEFOS (http://www.gefos.org/) (ebi-a-GCST005348,ieu-a-980,ieu-a-977,and ieu-a-982). Data sets in detail are available in the supplementary materials.

**Funding:** This study was supported by the Natural Science Foundation of Guangdong Province (2023A1515010425) and the Shenzhen Science and Technology Innovation grant (No. JCYJ20220531091200001). There was no additional external funding received for this study.

**Competing interests:** NO authors have competing interests.

**Abbreviations:** 25-OHD, 25-Hydroxy itamin D; ALD, alcoholic liver disease; BMD, bone mineral density; CI, confidence interval; DXA, dual-energy x-ray absorptiometry; FA, forearm; FN, femoral neck; GEFOS, genetic factors for osteoporosis consortium website; GWASs, genome-wide association studies; IV, instrumental variable; IVW, inverse variance-weighted; LD, linkage disequilibrium; LS, lumbar spine; MAF, Minor allele frequency; MR, mendelian randomization; MR-PRESSO, MR-pleiotropy residual sum and outlier; MVMR, Multivariate mendelian randomization; NAFLD, non-alcoholic fatty liver; OR, Odds ratio; RAPS, Robust adjusted profile score; SNP, single-nucleotide polymorphism; TB, total body.

## Results

The MR results suggested a robust genetically predicted effect of ALD on reduced BMD in the femoral neck (FN-BMD) (IVW beta = -0.0288; 95% CI: -0.0488, -0.00871; P = 0.00494) but not the other three sites. Serum 25-OHD level exhibited a significant mediating effect on the association between ALD and reduced FN-BMD albeit the proportion of mediation was mild (2.21%). No significant effects of NAFLD, alcohol consumption, or smoking on BMD in four sites, or reverse effect of BMD on ALD or NAFLD were detected.

## Conclusion

Our findings confirm the genetically predicted effect of ALD on reduced FN-BMD, and highlight the importance of periodic BMD and serum 25-OHD monitoring and vitamin D supplementation as needed in patients with ALD. Future research is required to validate our results and investigate the probable underlying mechanisms.

## Introduction

Osteoporosis, the most common bone condition characterized by low bone mineral density (BMD) and thereby increased bone fragility [1], affects approximately 500 million men and women worldwide [2]. Osteoporosis increases the risk of fracture, particularly hip fractures, and is the main cause of disability and death in the elderly population [3], posing an enormous threat to individuals, families, and society.

Overall, 60% of osteoporosis cases are primary osteoporosis resulting from the aging process, while 40% are secondary to chronic diseases, including chronic liver diseases. The liver is the metabolic center involving not only in the metabolism of micro- and macro-nutrients (glucose, fat, protein, vitamins) nourishing the bones but also the synthesis of a variety of signaling molecules such as cytokines and enzymes into the circulation to modulate the metabolism and function of the bones [4]. Patients with chronic liver disease may experience abnormal metabolic changes in bones, known as hepatic osteodystrophy commonly manifested as decreased BMD and osteoporosis [5, 6]. Alcoholic liver disease (ALD) and non-alcoholic fatty liver disease (NAFLD) are two major types of chronic liver diseases marked by a variety of abnormalities in the liver, including fatty liver disease, liver fibrosis, and cirrhosis [7]. Previous observational studies have demonstrated reduced BMD and increased susceptibility to intertrochanteric and hip fractures in patients with ALD [8]. In contrast, the correlation between NAFLD and osteoporosis remains controversial with both positive and null correlations having ever been reported previously [9, 10]. However, previous studies are largely based on an observational cross-sectional design which is susceptible to confounding bias and reverse causality, thus there is still a lack of genetic evidence confirming the causal relationship between ALD, NAFLD, and BMD. Given the dramatically increasing prevalence of ALD and NAFLD, particularly NAFLD, it is of great clinical significance to elucidate their causal relationship with BMD and identify potential mediators for osteoporosis prevention and management.

Mendelian randomization (MR) analysis is a powerful research method that evaluates the causal link between exposure and outcome by using one or more genetic variants [11]. MR studies can avoid the drawbacks of conventional observational investigations that are susceptible to confounding bias and reverse causality [12]. Multivariate MR (MVMR) is an extension of univariate MR that can be used to estimate the effect of multiple exposures on the outcome in a single model [13]. A two-step approach combining univariate MR and MVMR can be

applied as mediator analysis to identify potential mediators through estimating the total and direct effect of an exposure on an outcome [14]. This approach retains the advantages of MR analysis for causal inference, such as avoiding confounding bias and reverse causality, while allowing for mediator analysis that requires the estimation of different effects [14].

In this study, using univariate and multivariate MR approaches, we aimed to estimate the association between genetically predicted chronic liver diseases including ALD and NAFLD, and genetic susceptibility to BMD, and further estimate whether and to what extent this association is influenced by common risk factors associated with ALD and NAFLD, particularly vitamin D deficiency which has been suggested to influence BMD in previous observational studies [15].

## Materials and methods

### Data sources

**ALD and NAFLD.** The design of this study is illustrated in Fig 1. The GWAS summary statistics of ALD and NAFLD were obtained from the FinnGen Research Project (https://r5.finngen.fi/). After controlling for age, sex, genetic correlation, genotyping batches, and the first 10 major components, 1416 cases of ALD (n = 215,008) and 894 cases of NAFLD (n = 218,792) were included in our study (S1 Table of S1 File).

**BMD.** BMD was assessed by dual-energy X-ray absorptiometry (DXA). Total body BMD (TB-BMD) is the most appropriate method for unbiased assessment of BMD variation in the same skeletal site from children to older adults [16, 17], while the femoral neck (FN), lumbar spine (LS), and forearm (FA) are three most commonly measured sites for osteoporosis diagnosis in postmenopausal women and men. In this study, GWAS summary statistics for BMD (unit, g/cm2) were obtained from the GEnetic Factors for Osteoporosis Consortium website (GEFOS, http://www.gefos.org/). GWAS summary statistics of TB-BMD were obtained from a meta-analysis of 56,284 European participants [17]. Summary statistics of FN-BMD (n = 32,735), LS-BMD (n = 28,498), and FA-BMD (n = 8143) from European ancestry were obtained from a previous report, which has the biggest GWAS on DXA-measured BMD [16]. The corresponding effect estimates of SNP on BMD had been adjusted for many principal components as described in previous studies [16, 17].

### Potential mediators/risk factors

Smoking was studied as 'number of cigarettes per day'(mean = 15 cigarettes/day) in a GWAS of 33,299 participants of European ancestry in the UK Biobank [18].

Summary data on alcohol consumption per week (mean = 15.13 units per week) were obtained from a GWAS analysis of 112,117 individuals from the British sample of UK Biobank [19]. Summary data for serum 25-Hydroxyvitamin D levels (25-OHD) were obtained from a GWAS analysis of 417,580 participants of European ancestry in the UK-Biobank [20].

The sources of the summary data used in this study and the relevant cohort information are shown in S1 Table of S1 File.

### Genetic instrument selection

Three main assumptions must be met by the chosen instrumental variants (IVs): 1) it is connected to the exposure; 2) it is not connected to the outcome via a confounding pathway; and 3) it is not directly connected to the outcome but only be indirectly connected through the exposure. Usually, instrumental SNPs are selected with a threshold requirement of $P < 5 \times 10^{-8}$ at the genome-wide significance level. However, using this threshold, only one SNP associated

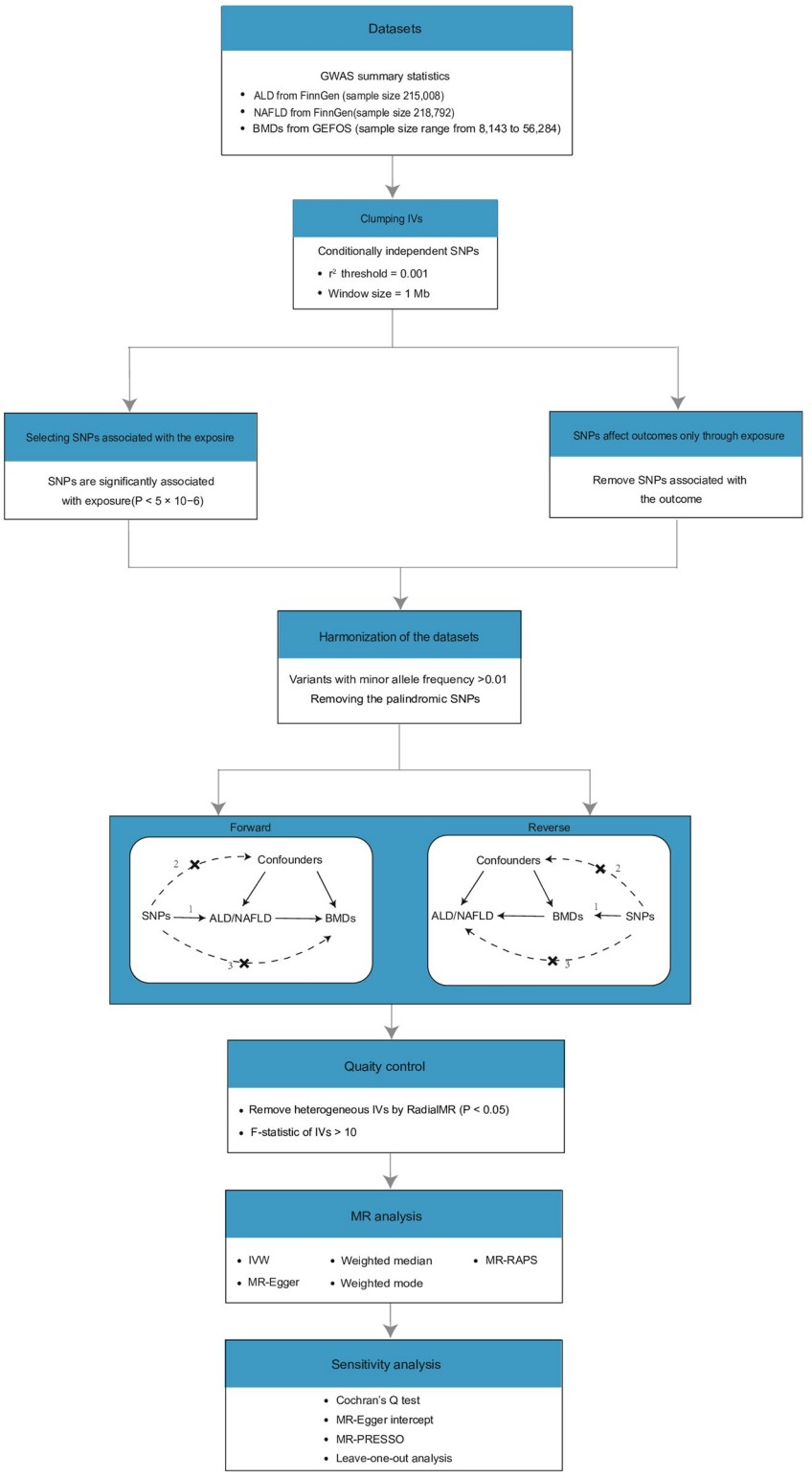

**Fig 1. Workflow of the causal inference between chronic liver disease (ALD and NAFLD) and BMDs.** GEFOS: GEnetic Factors for OSteoporosis Consortium website; ALD: alcoholic liver disease; BMD: bone mineral density; SNP: single nucleotide polymorphism; NAFLD: non-alcoholic fatty liver; IVs: instrumental variables; RAPS: robust adjusted profile score.

with ALD and two SNPs associated with NAFLD were obtained from FinnGen. Generally, at least 10 IVs are required for MR study [21, 22]; therefore, we applied a less strict criterion of P-value < 5 x10$^{-6}$ according to previous reports [22, 23]. We excluded outcome-related variants with P-values lower than nominal P-values with Bonferroni correction for the number of SNPs (P < 0.0500/Number of SNPs) to meet the assumption that instrumental variables are only associated with exposure outcomes [24]. Variants with MAF <1% in the GWAS datasets were excluded. Strand-ambiguous SNPs with intermediate allele frequencies in the harmonized exposure-outcome dataset were also removed. Outlier pleiotropic SNPs were detected using the "RadialMR" package [25]. The strength of the IVs can be quantified using the F-statistic ($F = beta^2/se^2$) [26]. An F-statistic greater than 10 reveals a low possibility of bias caused by weak instrumental variables in MR analysis [27]. In addition, we estimated the proportion of variance ($R^2$) explained by the association between genetic instruments and exposure variables. The following equation was used: $R^2 = 2 \times EAF \times (1—EAF) \times beta^2$, where EAF is the effect allele frequency, and beta is the estimated genetic effect on exposure [28]. In addition, we used the Power Calculator (https://sb452.shinyapps.io/power/) to determine whether our study had sufficient power to detect an association between exposure and outcome.

## Univariate MR

Univariate MR analysis was carried out to investigate the genetically predicted effect of ALD and NAFLD on BMD using the "TwoSampleMR" (V0.5.6) R package [29]. The random-effects inverse-variance weighted (IVW) method was used as the primary method of analysis [30]. This method performs a meta-analysis to determine the genetically predicted relationship between the exposure and the outcome by combining the estimates of the Wald ratio derived from each SNP. Considering the uncertainty associated with pleiotropy, the random-effects IVW approach obtains a more conservative inference of causality than the standard fixed-effects IVW [25]. Unless otherwise stated, all "IVW" here refers to the IVW for random effects. To avoid bias in the IVW results caused by the horizontal pleiotropy of any single SNP, four other MR methods were used to enhance the robustness of the results, including the MR-robust adjusted profile score (RAPS), MR-Egger, weighted median, and weighted mode approaches. By considering systematic and particular pleiotropy, the MR-RAPS approach allows for a robust inference for MR studies employing weak instruments [31]. Slope coefficients from MR-Egger regressions provide consistent estimates of causal effects in the presence of horizontal pleiotropy [32]. Even when 50.0% of the data contributing to the total estimate originates from incorrect genetic variations, the weighted median approach enables a consistant estimate [33].

When type I error rates are low and the modified IV assumptions exhibit less bias, the weighted mode technique yields consistent estimates [33, 34].

As the BMD of four different sites was considered, a Bonferroni correction was applied to counteract the problem of multiple comparisons (P = 0.05/4 = 0.0125) [35]. Therefore, P <0.0125 was considered statistically significant in this study.

## Sensitivity analysis

The IVW estimates may be biased when there is pleiotropy in the IVs. Therefore, we supplemented several sensitivity analyses to determine the causal effects [30]. First, Cochran's Q statistic was used to calculate the IVW estimate in a fixed-effects model and to check the heterogeneity [36, 37]. A random-effects IVW MR model should be utilized if Cochran's Q statistic is suggestive of potential pleiotropy. To detect possible bias in directional pleiotropy, MR-Egger regression was employed [32]. The average pleiotropic effect of all genetic

variations was used to represent the MR-Egger regression's intercept term. When this value was not equal to zero (P <0.05), evidence of pleiotropy was suggested. Next, the Mendelian Randomization Pleiotropy RESidual Sum and Outlier (MR-PRESSO) test was performed to investigate the presence of horizontal pleiotropy [38] using the "MR-PRESSO" R package (https://github.com/rondolab/MR-PRESSO/). Furthermore, a leave-one-out analysis was performed to test whether individual SNPs significantly affected the estimate of the causal effect. We further performed the MR-Steiger test to determine the direction of causality of the association in the univariate MR [39].

## Multivariate MR and mediation analysis

Multivariate MR was used to estimate the independent effect of chronic liver disease (ALD and NAFLD) on BMD, controlling for common risk factors such as smoking, alcohol intake, and vitamin D deficiency [40]. In addition, we further explored the extent to which chronic liver disease affects BMD through potential mediators using two-step MR [41, 42], and calculated the mediating effect using the coefficient product method. Firstly, the total effect of chronic liver disease on the risk of BMD was estimated by using the univariate MR (coefficient beta XY) and estimating the effect of chronic liver disease on the potential mediator (coefficient beta XM). Here, X, M, and Y represent exposure, mediator, and outcome, respectively. Secondly, multivariate MR was used to estimate the direct effect of the potential mediator on BMD (coefficient betaMY). We divided the indirect effect (betaXM × betaMY) by the total effect (betaXY) to obtain the proportion of the mediating effect. It is worth noting that only those chronic liver disease or potential mediators that were statistically significant in univariate MR were included in the MR-based mediator analysis.

## Bidirectional MR

The MR analysis described above was extended to a bidirectional causal inference between chronic liver disease and BMD. MR analysis was performed, with ALD or NAFLD as exposure and BMD as the outcome. Conversely, a reverse MR analysis was performed, with BMD as exposure and ALD or NAFLD as the outcome of the study.

We followed the STROBE-MR checklist [43] and the guidelines of Burgess et al. for conducting MR analysis [44].

## Ethics statement

Our analysis used publicly available GWAS summary data. No original data was collected for this manuscript; therefore, no ethical committee approval was required. Each study included was approved by their institutional ethics review committees, and all participants provided written informed consent.

## Results

### Genetic instruments

For forward MR, we used genetic instruments with 15, 12, 20, 45, and 223 conditionally independent SNPs for ALD, NAFLD, number of cigarettes per day, alcohol consumption per week, and serum 25-OHD levels, respectively (S2-S6 Tables of S1 File). For reverse MR, we used a genetic tool with 155, 60, 54, and 16 conditionally independent SNPs for TB-, FA-, FN-, and LS-BMD, respectively (S7-S10 Tables of S1 File). The F-statistics for all SNPs ranged from 20 to 1475, indicating that these instrumental variables are unlikely to have a weak instrumental bias (S2-S10 Tables of S1 File). Outliers SNPs detected by the "RadialMR" package were

removed (S11 Table of S1 File, S1–S4 Figs). No SNPs for exposure were strongly correlated with the outcome of our study. After filtering for SNPs as described above, the remaining SNPs were used for univariate MR analysis (S12-S19 Tables of S1 File).

## Estimating the effect of ALD, NAFLD, and potential mediators on BMD

As shown in Fig 2, genetically determined ALD was significantly associated with a decreasing trend of FN-BMD at the Bonferroni corrected significance of P < 0.0125 (IVW beta = -0.0288; 95% CI: -0.0488, -0.00871; P = 0.00494). The results of the MR-Egger (beta = -0.00822; 95% CI: -0.0464, 0.0300; P = 0.682), weighted median (beta = -0.0200; 95% CI: -0.0473, 0.00732; P = 0.151), weighted mode (beta = -0.0155; 95% CI: -0.0485, 0.0175; P = 0.377) and MR-RAPS (beta = -0.0282; 95%CI: -0.0508, 0.00563; P = 0.0143) yielded similar trends, although the P values did not reach statistical significance (Fig 2). Notably, with the current sample size, our study has sufficient power (>80%) to detect an association between ALD and FN-BMD given the 34.48% phenotypic variation in ALD (S20 Table of S1 File). However, ALD had no significant effect on TB-BMD, LS-BMD, or FA-BMD (Fig 2).

No significant effects of NAFLD on TB-BMD, FN-BMD, LS-BMD, and FA-BMD were observed (Fig 3).

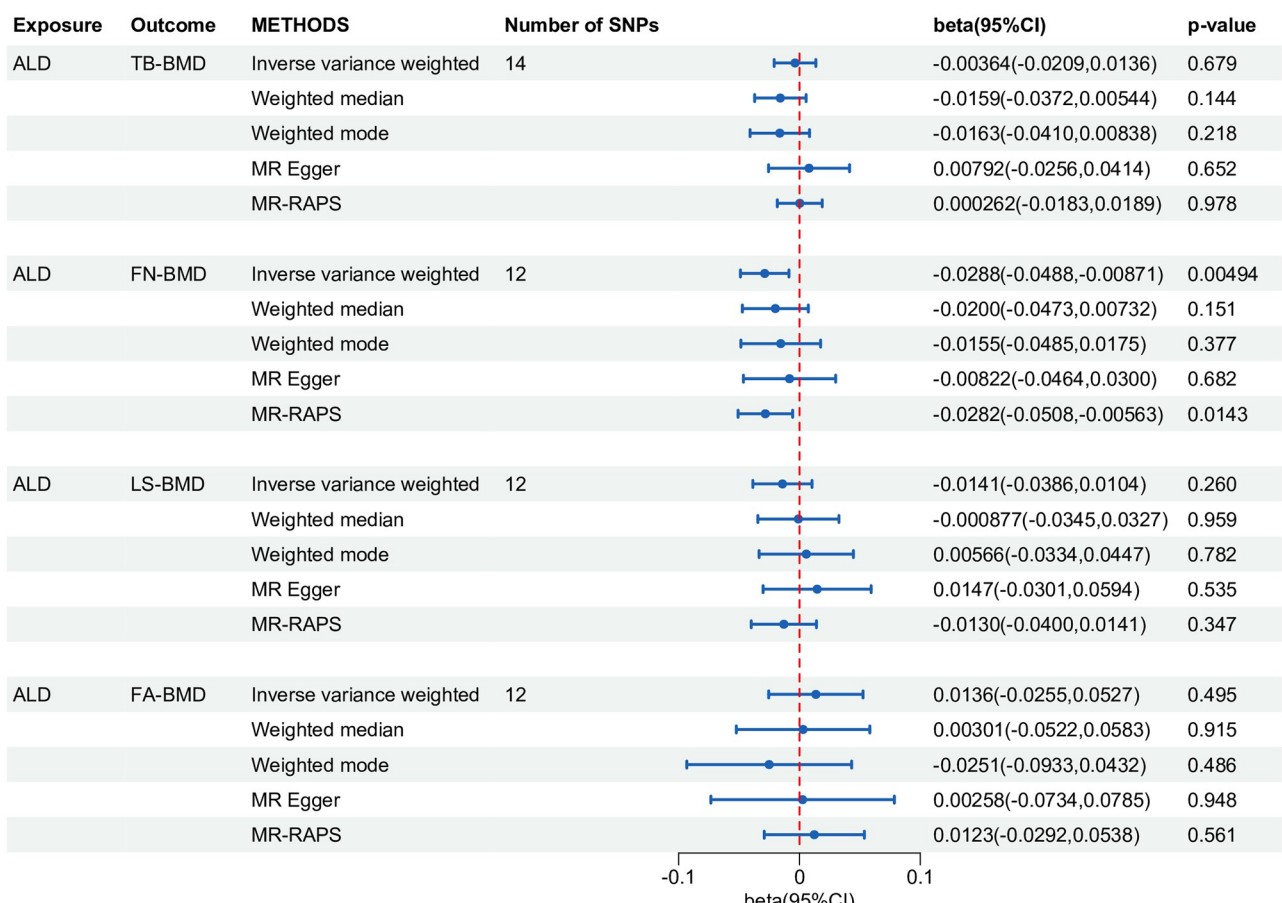

| Exposure | Outcome | METHODS | Number of SNPs | | beta(95%CI) | p-value |
|---|---|---|---|---|---|---|
| ALD | TB-BMD | Inverse variance weighted | 14 | | -0.00364(-0.0209,0.0136) | 0.679 |
| | | Weighted median | | | -0.0159(-0.0372,0.00544) | 0.144 |
| | | Weighted mode | | | -0.0163(-0.0410,0.00838) | 0.218 |
| | | MR Egger | | | 0.00792(-0.0256,0.0414) | 0.652 |
| | | MR-RAPS | | | 0.000262(-0.0183,0.0189) | 0.978 |
| ALD | FN-BMD | Inverse variance weighted | 12 | | -0.0288(-0.0488,-0.00871) | 0.00494 |
| | | Weighted median | | | -0.0200(-0.0473,0.00732) | 0.151 |
| | | Weighted mode | | | -0.0155(-0.0485,0.0175) | 0.377 |
| | | MR Egger | | | -0.00822(-0.0464,0.0300) | 0.682 |
| | | MR-RAPS | | | -0.0282(-0.0508,-0.00563) | 0.0143 |
| ALD | LS-BMD | Inverse variance weighted | 12 | | -0.0141(-0.0386,0.0104) | 0.260 |
| | | Weighted median | | | -0.000877(-0.0345,0.0327) | 0.959 |
| | | Weighted mode | | | 0.00566(-0.0334,0.0447) | 0.782 |
| | | MR Egger | | | 0.0147(-0.0301,0.0594) | 0.535 |
| | | MR-RAPS | | | -0.0130(-0.0400,0.0141) | 0.347 |
| ALD | FA-BMD | Inverse variance weighted | 12 | | 0.0136(-0.0255,0.0527) | 0.495 |
| | | Weighted median | | | 0.00301(-0.0522,0.0583) | 0.915 |
| | | Weighted mode | | | -0.0251(-0.0933,0.0432) | 0.486 |
| | | MR Egger | | | 0.00258(-0.0734,0.0785) | 0.948 |
| | | MR-RAPS | | | 0.0123(-0.0292,0.0538) | 0.561 |

**Fig 2. Forest plot of univariate MR analysis estimating the effect of ALD on BMDs.** MR: mendelian randomization; ALD: alcoholic liver disease; BMD: bone mineral density; SNP: single nucleotide polymorphism; RAPS: robust adjusted profile score; TB: total body; FN: femoral neck; LS: lumbar spine; FA: forearm; CI: confidence interval.

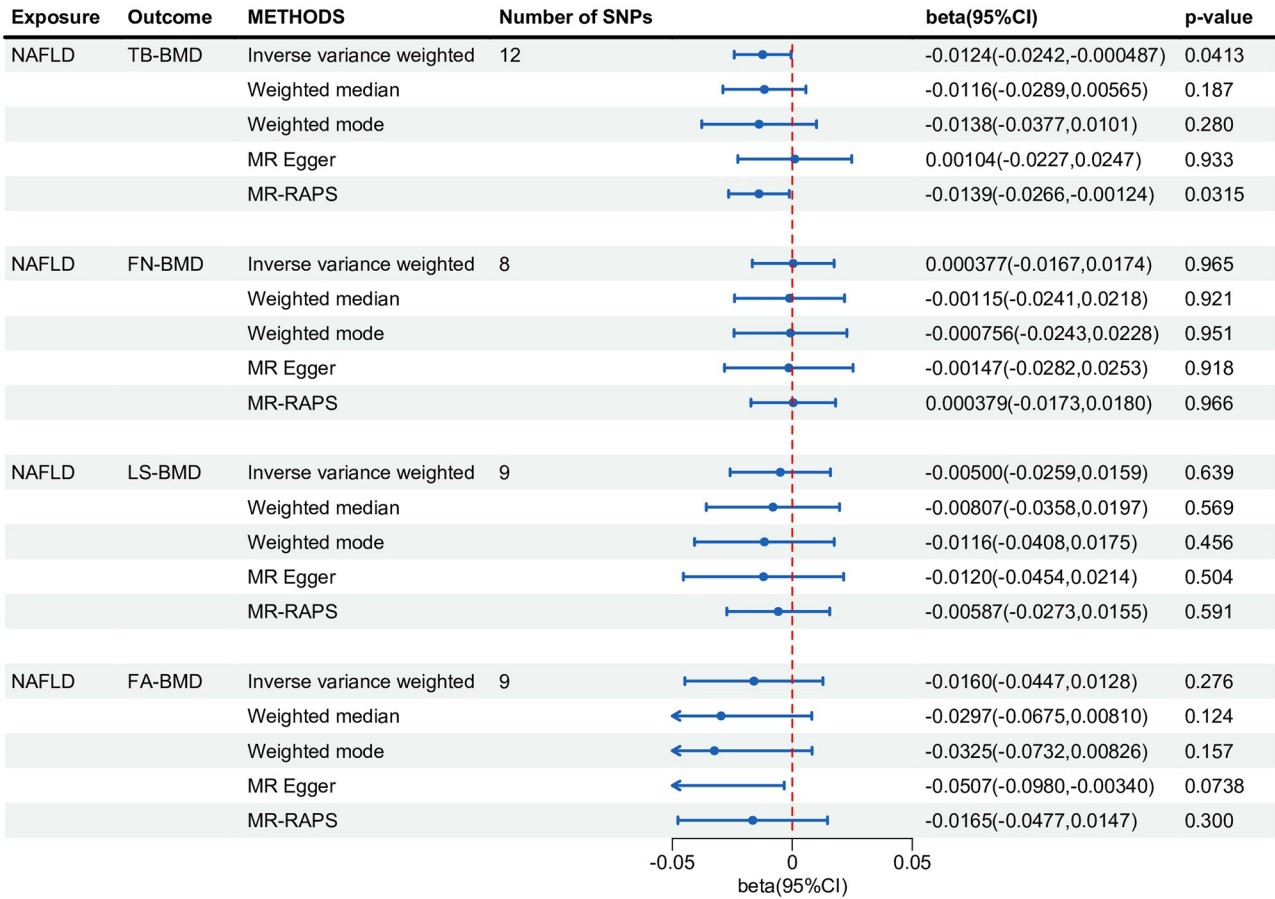

**Fig 3. Forest plot of univariate MR analysis estimating the effect of NAFLD on BMDs.** MR: mendelian randomization; BMD: bone mineral density; SNP: single nucleotide polymorphism; NAFLD: non-alcoholic fatty liver; RAPS: robust adjusted profile score; TB: total body; FN: femoral neck; LS: lumbar spine; FA: forearm; CI: confidence interval.

Genetically determined serum 25-OHD levels were associated with an increasing trend in FN-BMD (beta = 0.126; 95%CI: 0.0512,0.201; P = 0.000976) (S21 Table of S1 File), but not in TB-BMD, LS-BMD, or FA-BMD (S21 Table of S1 File). However, the effect of number of cigarettes per day and alcohol consumption per week on TB-BMD, FN-BMD, LS-BMD, and FA-BMD did not reach statistical significance (S22, S23 Tables of S1 File).

## Evaluating the direct and indirect effects of ALD on BMD risk

The genetic instrumental variables used for multivariate MR are shown in S24-S27 Tables of S1 File. After adjusting for number of cigarettes per day and alcohol consumption per week, the genetically predicted effect of ALD on FN-BMD remained significant (all P < 0.0125) (Table 1). However, after adjusting for serum 25-OHD levels, the effect of ALD on FN-BMD was attenuated toward null (beta = -0.0122; 95%CI: -0.0324, 0.00803; P = 0.237) (Table 1). Interestingly, the direct effect of serum 25-OHD levels on FN-BMD was significant after adjustment for ALD (beta = 0.0965; 95%CI: 0.0302,0.163; P = 0.00433) (Table 1). This effect remained significant with further controlling for number of cigarettes per day and alcohol consumption per week (beta = 0.0922; 95%CI: 0.0211, 0.163; P = 0.0110) (Table 1).

**Table 1. Multivariable MR analysis estimating the direct and indirect effect of ALD on FN-BMD, conditioning on number of cigarettes per day, alcohol consumption per week, and serum 25-OHD levels.**

| Models | Exposures | No. of SNPs | beta | 95% CI | pval |
|---|---|---|---|---|---|
| Model 1 | | | | | |
| | alcoholic liver disease | 31 | -0.0299 | (-0.0505,-0.00943) | 0.00422 |
| | number of cigarettes per day | | -0.0452 | (-0.160,0.0698) | 0.441 |
| Model 2 | | | | | |
| | alcoholic liver disease | 51 | -0.0295 | (-0.0505,-0.00857) | 0.00572 |
| | alcohol consumption per week | | -0.0642 | (-0.280,0.151) | 0.560 |
| Model 3 | | | | | |
| | alcoholic liver disease | 202 | -0.0122 | (-0.0324,0.00803) | 0.237 |
| | serum 25-OHD levels | | 0.0965 | (0.0302,0.163) | 0.00433 |
| Model 4 | | | | | |
| | alcoholic liver disease | 209 | -0.0143 | (-0.0352,0.00652) | 0.178 |
| | number of cigarettes per day | | -0.0833 | (-0.202,0.0354) | 0.169 |
| | alcohol consumption per week | | -0.153 | (-0.399,0.0945) | 0.226 |
| | serum 25-OHD levels | | 0.0922 | (0.0211,0.163) | 0.0110 |

MR: mendelian randomization; SNP: single nucleotide polymorphism; CI: confidence interval;25-OHD: 25-Hydroxyvitamin D.

## Serum 25-OHD levels mediate the genetically predicted effect of ALD on FN-BMD

We investigated the role of serum 25-OHD levels as mediators in the association between ALD and FN-BMD by MR-based mediation analysis. Firstly, we found a negative association between genetically determined ALD and serum 25-OHD levels using univariate MR (beta = -0.00661; 95%CI: -0.0114, -0.00183; P = 0.00672) (Fig 4, S28 Table of S1 File). Next,

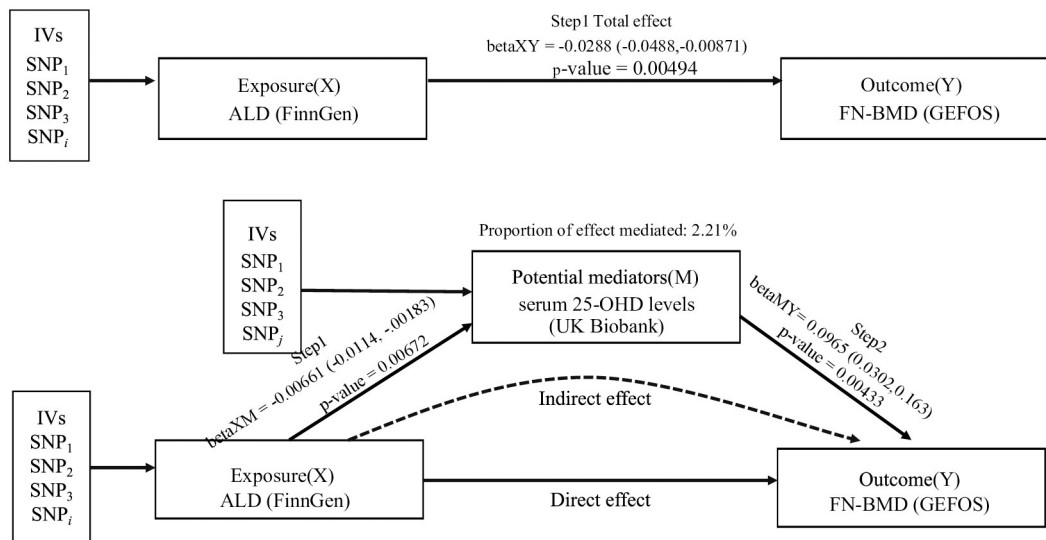

**Fig 4. Mediation analysis of the effect of ALD on FN-BMD via serum 25-OHD.** Step1:the total effect of ALD on the risk of FN-BMD was estimated by using the univariate MR (coefficient betaXY) and estimating the effect of ALD on serum 25-OHD (coefficient beta XM);Step 2: multivariate MR was used to estimate the direct effect of serum 25-OHD on FN-BMD (coefficient betaMY). ALD: alcoholic liver disease; FN: femoral neck; BMD: bone mineral density; CI: confidence interval; 25-OHD:25-Hydroxyvitamin D; GEFOS: GEnetic Factors for Osteoporosis Consortium.

multivariate MR was used to estimate the direct effect of serum 25-OHD levels on FN-BMD. Genetically determined serum 25-OHD levels are associated with increased risk of FN-BMD after adjustment for ALD (beta = 0.0965; 95%CI: 0.0302,0.163; P = 0.00433)(Fig 4). The proportion of the total genetically predicted effect of ALD on FN-BMD risk mediated by serum 25-OHD level was estimated to be 2.21% (Fig 4).

### Reverse Mendelian randomization analysis of BMD on ALD and NAFLD

The IVW estimates suggest null effect of TB-BMD (IVW OR = 0.972; 95%CI: 0.828,1.14; P = 0.727), FN-BMD (IVW OR = 1.06; 95% CI: 0.846,1.32; P = 0.623), LS-BMD (IVW OR = 1.02; 95% CI: 0.831,1.25; P = 0.843) or FA-BMD (IVW OR = 0.875; 95% CI: 0.712,1.08; P = 0.202) on ALD (S29 Table of S1 File). Similar findings were found for the effects of TB-BMD (IVW OR = 1.16; 95%Ci: 0.941,1.42; P = 0.165), FN-BMD (IVW OR = 1.19; 95%CI: 0.903, 1.58; P = 0.215), LS-BMD (IVW OR = 1.20; 95%CI: 0.922,1.57; P = 0.173) or FA-BMD (IVW OR = 1.24; 95%CI: 0.954,1.60; P = 0.469) on NAFLD(S30 Table of S1 File). The MR-Egger, weighted median, weight mode, and MR-RAPS approaches also provided consistent effect estimates (S29, S30 Tables of S1 File).

### Sensitivity analyses

Evidence of horizontal polymorphism (MR-Egger intercept P-value = 0.0367) was found in analyses where exposure was weekly alcohol consumption and the outcome was LS-BMD (S23 Table of S1 File). Otherwise, no evidence of horizontal pleiotropy (MR-Egger intercept all P-value >0.05) was detected in the other groups (Table 2; S21-S23, S28-S30 Tables of S1 File). Additionally, the MR-PRESSO global tests detected no significant pleiotropy and the Cochran's Q tests detected no significant heterogeneity (Table 2; S21-S23, S28-S30 Tables of S1 File).

The scatter plot depicted the relative risk associations of each SNP with exposures and outcomes (S5–S8 Figs). Leave-one-out analysis showed that no single SNP droved these results (S9–S14 Figs). We detected no evidence of reverse causality with the "MR Steiger" test (S31 Table of S1 File).

### Discussion

In this study, we used an MR approach to investigate the causal relationship between ALD, NAFLD, and BMD, and their potential mediators. Our study demonstrated that genetically

**Table 2. Horizontal pleiotropy and heterogeneity of results.**

| Exposures | Outcomes | Heterogeneity Test | | | | Pleiotropy test | | | |
|---|---|---|---|---|---|---|---|---|---|
| | | IVW | | MR-Egger | | MR-Egger intercept | | | MR-PRESSO |
| | | Q-statistics | P | Q-statistics | P | Estimate | SE | P | Global test P |
| ALD | TB-BMD | 18.482 | 0.140 | 17.565 | 0.130 | -0.00525 | 0.00662 | 0.444 | 0.162 |
| | FN-BMD | 9.554 | 0.571 | 8.019 | 0.627 | -0.0108 | 0.00869 | 0.244 | 0.614 |
| | LS-BMD | 12.102 | 0.356 | 9.924 | 0.447 | -0.0150 | 0.0101 | 0.171 | 0.355 |
| | FA-BMD | 8.721 | 0.648 | 8.611 | 0.569 | 0.00583 | 0.0176 | 0.747 | 0.609 |
| NAFLD | TB-BMD | 10.668 | 0.471 | 9.029 | 0.529 | -0.00790 | 0.00619 | 0.229 | 0.496 |
| | FN-BMD | 2.534 | 0.925 | 2.503 | 0.868 | 0.00165 | 0.00939 | 0.867 | 0.944 |
| | LS-BMD | 9.445 | 0.306 | 9.061 | 0.248 | 0.00518 | 0.00952 | 0.603 | 0.381 |
| | FA-BMD | 6.225 | 0.622 | 2.945 | 0.890 | 0.0254 | 0.0140 | 0.113 | 0.634 |

ALD: alcoholic liver disease; BMD: bone mineral density; TB: total body; FN: femoral neck; LS: lumbar spine; FA: forearm; NAFLD: non-alcoholic fatty liver.

predicted ALD was associated with decreased FN-BMD, which was unlikely to be influenced by smoking and alcohol consumption. Mediation analysis showed that the effect of ALD on FN-BMD was partly mediated by serum 25-OHD levels. These findings suggest that excessive alcohol intake does not directly affect the alteration of BMD; however, if alcohol over-intake leads to liver damage and ALD, it can lead to reduced BMD, partially through vitamin D insufficiency resulting from ALD. Therefore, our study reinforces the importance of BMD and serum 25-OHD level monitoring and vitamin D supplement as needed in patients with ALD.

Our findings regarding the causal relationship between ALD and low FN-BMD corroborate previous epidemiological studies [8]. Hip fractures occurred more than five times as frequently in patients with alcoholic cirrhosis than in their counterparts, according to a prior large cohort study in a European population [45]. Consistently, patients with alcoholic liver cirrhosis had considerably poorer measurements of bone strength, including BMD, in their proximal femurs compared to healthy controls [46]. Similarly, another observational investigation found that FN-BMD declined in ALD donors compared to healthy controls, with the intertrochanteric region showing the most noticeable bone strength decrease [8, 36]. Therefore, together with previous epidemiological studies, our findings highlight the importance of monitoring FN-BMD for hip fracture prevention in patients with ALD.

The biological mechanisms underlying the causal link between ALD and low FN-BMD remain unclear. We suspect that the lower FN-BMD might be partially, if not fully, attributable to metabolic dysregulation of the liver. Of note, vitamin D insufficiency and deficiency may play a critical role in bone mass loss and decreased BMD secondary to ALD. According to a previous study, vitamin D inadequacy was established in 90% of patients with alcoholic liver cirrhosis [47]. Vitamin D is essential for calcium absorption, bone matrix formation, and bone mineralization which is positively associated with BMD [48]. The liver is the main place of vitamin D 25-hydroxylation which is a critical process of vitamin D activation [49]. Consistent with previous knowledge, we demonstrated that the genetically predicted effect of ALD on reduced FN-BMD is mediated by serum 25-OHD levels. Therefore, our study highlights the importance of periodically evaluating vitamin D status and taking vitamin D supplements as needed in patients with ALD to reduce the risk of decreased BMD and osteoporosis. Apart from vitamin insufficiency, other mechanisms may also be involved in this process. For instance, chronic liver inflammation is associated with decreased absorption of nutrients from the gastrointestinal tract and decreased protein synthesis, resulting in bone nutrition deficiency [50]. Insulin-like growth factor-1 deficiency, which has been associated with liver diseases of several etiologies, including ALD, contributes to impaired mineralization, osteoblast proliferation, and collagen synthesis in the bone matrix [51]. Altogether, our findings highlight the genetically predicted link between ALD and reduced FN-BMD, although the detailed mechanisms require further investigation.

In this study, we demonstrated that the impact of ALD on BMD was different at different skeletal sites, with a significant causal effect on FN-BMD but not on TB-BMD, LS-BMD, or FA-BMD. The mechanisms underlying these different effects at different sites remain unclear. We suspect that the difference may be related to the anatomy of the bones. Unlike the LS and FA, the FN is located at the junction of the blood supply from two terminals. The vessels of the femoral neck are relatively smaller with lower blood flow rates than those at other sites. Therefore, the femoral neck may be more susceptible to impaired blood supply and nutritional deficiencies as a result of ALD. In support of our hypothesis, lower bone flow in the hip joint than in the spine was confirmed by fluorodeoxyglucose positron emission tomography in a previous study [52]. Nevertheless, the statistical power of other sites is relatively lower than that of

FN-BMD; therefore, future studies with larger statistical power are needed to confirm our findings.

Until now, the effect of NAFLD on BMD remains controversial. While some studies reported a deteriorating impact of NAFLD on BMD, others argued that there is no correlation between them [9, 10]. In consistency, the impact of obesity, the major cause of NAFLD, on BMD is also complex. On one hand, obesity may lead to increased BMD because it is associated with higher mechanical load, higher 17 β-estradiol levels, and better nutrition status [53]. On the other hand, obesity is associated with low-grade chronic inflammation and increased inflammatory cytokines concentrations, which may contribute to accelerated bone loss in obesity [53]. In this study, we found no significant association between genetically predicted NAFLD and BMD. We suspected that the effect of NAFLD on BMD and osteoporosis might be dependent on the severity and disease duration of NAFLD, therefore we were not able to detect a significant effect without considering NALFD severity. In support of our hypothesis, some studies demonstrated that hepatic fibrosis may be more closely related to osteoporosis than simple steatosis [54–57]. However, our findings should be interpreted with caution due to limited power, and further studies are warranted to confirm our findings.

## Strengths and limitations

Our research has several advantages. First, residual confounding and reverse causation, which are common in traditional observational studies, were greatly decreased in this MR research design by using genetic variants as instrumental variables. In addition, the current study utilized the largest sample of GWAS summary statistics to date for analysis. Besides, we used a series of sensitivity analyses to make our results more robust. Moreover, we used an MVMR design to address the effect of alcohol and tobacco consumption and identify potential mediator serum 25-OHD level.

However, this study has some limitations that deserve attention. First, regarding the selection of genetic instruments, only one SNP was identified for association with ALD and two SNPs for association with NAFLD if a conventional $P < 5 \times 10^{-8}$ were applied. However, a minimum of 10 instrumental variables is required to conduct MR studies [21, 22], and too few SNPs would make it difficult to match instrumental variables in the outcomes. For this reason, we chose SNPs with a less stringent significance threshold of $5 \times 10^{-6}$, as described in previous studies [22, 23]. However, the use of a less strict P value did not bring weak instrument bias; all the F statistics of these instrument variables were greater than 10 in the current study. Second, despite the selection of strongly correlated SNPs, these SNPs only explain part of the total variation in complex traits [58], and thus cannot be considered an exact proxy for the exposure. Third, without thorough functional tracing of these motifs, it is impossible to entirely rule out pleiotropic pathways since we do not yet fully understand the biological roles of these SNPs. However, we used a series of sensitivity analyses, including Cochran's Q statistic, MR-Egger regression, and MR-PRESSO to test for horizontal pleiotropy and heterogeneity, which were found unlikely to bias the findings in our study [59]. Fourth, the GWAS-pooled data we used were related only to European ancestry. Therefore, our findings might not accurately reflect the entire population; caution should be exercised when applying our findings to racially and ethnically diverse populations. Fifth, although we have used the most up-to-date GWAS datasets with largest sample size, the sample size is still limited for some variables, eg. FA-BMD. Finally, although we did not detect significant associations between ALD and BMD of other sites, nor the correlations between NAFLD and BMD, we could not rule out the possibility that it might be due to the limited statistical power of these analyses. Therefore, further studies with larger sample size and larger statistical power are warranted to confirm our findings.

## Conclusion

In conclusion, this study provides robust genetic evidence that ALD is associated with reduced FN-BMD, partially mediated by low serum 25-OHD levels secondary to ALD. Considering the tremendous burden of hip fractures on individuals and families, our findings reinforce the significance of FN-BMD and serum 25-OHD monitoring and vitamin D supplementation as needed for hip fracture prevention in patients with ALD.

## Supporting information

**S1 Fig. Result for the RadialMR analysis in the SNPs (RadialMR plot).** ALD (exposure) with A) TB-BMD(outcome); B) FN-BMD(outcome); C)LS-BMD(outcome); D) FA-BMD(outcome); NAFLD (exposure) with E) TB-BMD(outcome); F) FN-BMD(outcome); G)LS-BMD (outcome); H) FA-BMD(outcome).
(PDF)

**S2 Fig. Result for the RadialMR analysis in the SNPs (RadialMR plot).** number of cigarettes per day(exposure) with A)TB-BMD(outcome); B) FN-BMD(outcome); C)LS-BMD(outcome); D) FA-BMD(outcome); alcohol consumption per week(exposure) with E) TB-BMD(outcome); F) FN-BMD(outcome); G)LS-BMD(outcome); H) FA-BMD(outcome); Serum 25-Hydroxyvitamin D levels(exposure) with I) TB-BMD(outcome); J) FN-BMD(outcome); K) LS-BMD(outcome); L) FA-BMD(outcome).
(PDF)

**S3 Fig. Result for the RadialMR analysis in the SNPs (RadialMR plot).** ALD (exposure) with Serum 25-Hydroxyvitamin D levels (outcome).
(PDF)

**S4 Fig. Result for the RadialMR analysis in the SNPs (RadialMR plot).** TB-BMD(exposure) with A) ALD(outcome); B)NAFL D(outcome); FN-BMD(exposure) with C) ALD(outcome); D)NAFLD(outcome);LS-BMD(exposure)with E)ALD(outcome); F)NAFLD(outcome); FA-BMD(exposure) with G) ALD(outcome); H)NAFLD(outcome).
(PDF)

**S5 Fig. Scatterplot of associations of genetically predicted ALD and NAFLD on the risk of BMDs.** Scatterplot of associations of genetically predicted ALD on the risk of A)TB-BMD; B) FN-BMD;C) LS-BMD; D)FA-BMD; Scatterplot of associations of genetically predicted NAFLD on the risk of E)TB-BMD; F)FN-BMD;G) LS-BMD;H)FA-BMD.
(PDF)

**S6 Fig. Scatterplot of associations of genetically predicted number of cigarettes per day, alcohol consumption per week, and serum 25-Hydroxyvitamin D levels on the risk of BMDs.** Scatterplot of associations of genetically predicted number of cigarettes per day on the risk of A)TB-BMD; B)FN-BMD;C) LS-BMD; D)FA-BMD; Scatterplot of associations of genetically predicted of alcohol consumption per week on the risk of E)TB-BMD; F)FN-BMD;G) LS-BMD; H)FA-BMD; Scatterplot of associations of genetically predicted of serum 25-Hydroxyvitamin D levels on the risk of I)TB-BMD; J)FN-BMD;K) LS-BMD; L)FA-BMD.
(PDF)

**S7 Fig. Scatterplot of associations of genetically predicted of ALD on the risk of serum 25-Hydroxyvitamin D levels.**
(PDF)

**S8 Fig. Scatterplot of associations of genetically predicted of BMDs on the risk of ALD and NAFLD.** Scatterplot of associations of genetically predicted of TB-BMD on the risk of A)ALD; B) NAFLD; Scatterplot of associations of genetically predicted of FN-BMD on the risk of C) ALD;D) NAFLD; Scatterplot of associations of genetically predicted of LS-BMD on the risk of E)ALD;F) NAFLD; Scatterplot of associations of genetically predicted of FA-BMD on the risk of G)ALD;H) NAFLD.
(PDF)

**S9 Fig. Leave-one-out analysis of ALD and NAFLD on BMDs.** Leave-one-out analysis of ALD on A) TB-BMD; B) FN-BMD; C) LS = BMD; D) FA-BMD; Leave-one-out analysis of NAFLD on E) TB-BMD; F) FN-BMD; G) LS = BMD; H) FA-BMD.
(PDF)

**S10 Fig. Leave-one-out analysis of number of cigarettes per day, and alcohol consumption per week on BMDs.** Leave-one-out analysis of number of cigarettes per day on A) TB-BMD; B) FN-BMD; C) LS-BMD; D) FA-BMD; Leave-one-out analysis of alcohol consumption per week on E) TB-BMD; F) FN-BMD; G) LS-BMD; H) FA-BMD.
(PDF)

**S11 Fig. Leave-one-out analysis of serum 25-Hydroxyvitamin D levels on BMDs.** Leave-one-out analysis of serum 25-Hydroxyvitamin D levels on A) TB-BMD; B) FN-BMD; C) LS-BMD; D) FA-BMD.
(PDF)

**S12 Fig. Leave-one-out analysis of ALD on serum 25-Hydroxyvitamin D levels.**
(PDF)

**S13 Fig. Leave-one-out analysis of BMDs on ALD.** Leave-one-out analysis of A) TB-BMD; B) FN-BMD; C) LS-BMD; D) FA-BMD on ALD.
(PDF)

**S14 Fig. Leave-one-out analysis of BMDs on NAFLD.** Leave-one-out analysis of A) TB-BMD; B) FN-BMD; C) LS-BMD; D) FA-BMD on NAFLD.
(PDF)

**S1 File.**
(XLSX)

## Acknowledgments

We acknowledge the participants and investigators of the FinnGen study, the UK Biobank, and GEFOS.

## Author Contributions

**Conceptualization:** Qinyao Huang, Yuying Zhang.

**Data curation:** Qinyao Huang, Jianglong Guo.

**Formal analysis:** Jianglong Guo, Hongjun Zhao, Yuying Zhang.

**Funding acquisition:** Yuying Zhang.

**Investigation:** Yi Zheng.

**Project administration:** Yi Zheng, Yuying Zhang.

**Resources:** Hongjun Zhao, Yuying Zhang.

**Software:** Qinyao Huang, Jianglong Guo.

**Supervision:** Hongjun Zhao, Yuying Zhang.

**Validation:** Qinyao Huang, Jianglong Guo.

**Visualization:** Qinyao Huang.

**Writing – original draft:** Qinyao Huang.

**Writing – review & editing:** Jianglong Guo, Yi Zheng, Yuying Zhang.

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
