## [Decision Letter · Decision Letter 0]

28 May 2023

PONE-D-23-08785Causal Associations between Alcoholic Liver Disease and Bone Mineral Density: A bidirectional Mendelian randomization studyPLOS ONE

Dear Dr. Zhang,

Thank you for submitting your manuscript to PLOS ONE. After careful consideration, we feel that it has merit but does not fully meet PLOS ONE’s publication criteria as it currently stands. Therefore, we invite you to submit a revised version of the manuscript that addresses the points raised during the review process.

We look forward to receiving your revised manuscript.

Kind regards,

Chunyu Liu, PhD

Academic Editor

PLOS ONE

Journal Requirements:

“This study was supported by the Shenzhen Science and Technology Innovation grant (No. JCYJ20220531091200001 to YZ)(funder website：http://stic.sz.gov.cn/). The funders had no role in study design, data collection and analysis, decision to publish, or preparation of the manuscript.”

Reviewers' comments:

Reviewer's Responses to Questions

**Comments to the Author**

1. Is the manuscript technically sound, and do the data support the conclusions?

Reviewer #1: Partly

Reviewer #2: Partly

2. Has the statistical analysis been performed appropriately and rigorously? 

Reviewer #1: Yes

Reviewer #2: Yes

3. Have the authors made all data underlying the findings in their manuscript fully available?

Reviewer #1: Yes

Reviewer #2: Yes

4. Is the manuscript presented in an intelligible fashion and written in standard English?

Reviewer #1: Yes

Reviewer #2: Yes

5. Review Comments to the Author

Reviewer #1: The authors use a bidirectional two-sample Mendelian randomization to infer causal association between ALD and BMDs. Through these procedures, they infer a causal effect of genetically predicted ALD on reduced FN-BMD.

The analyses are interesting but further analyses are required to confirm validity of the finding.

1. The authors found that genetically determined ALD was associated with a decreasing trend of

FN-BMD using the IVW (β = -0.03, 95% CI: -0.05 to -0.01, P = 0.0077). Although the p-value is small, it is better to specify a threshold for multiple testing.

2. If multiple testing threshold was applied, then only the IVW method gave significant result, all the sensitivity analyses (4 tests) showed no causal association. In this case, is it still reasonable to conclude ALD was causally associated with FN-BMD? How would you explain the different results between IVW and those sensitivity analyses?

3. In supplemental table 2, rs75978489 has EAF 0. Is this a typo or a rare variants? All of the variants in this table have remarkably high R squared. How did the R squared calculated?

4. In results part, the author wrote “according to our selection criteria, 15 SNPs were maintained as IVs for ALD”, while figure A and B only showed 11 SNPs?

5. Given the null association for other BMDs, power calculations should be included to assess for risk of type 2 error

6. The authors state that there is evidence for a causal effect between ALD and FN-BMD. Could this be further evaluated –eg is there evidence that liver disease treatment (or liver transplant), or people who quit drinking associated with a rise of FN-BMD?

7. In discussion the fifth point, the authors state they cannot estimate the extent of overlap in the study. Could this be evaluated by checking the cohorts that each database included?

Minor:

1. I would tone down the claims for causality from the observational data throughout. Significant genetic associations are consistent with causality but one cannot definitively claim causality as the authors imply throughout.

2. Figure 1: Romoving should be removing

3. Page 15: weighted model -> weighted mode

4. Results: 138 SNPs for TB-BMD, while in supplemental table 3, 139 SNPs

5. In strengths and limitations: excessive period after “variations as IVs”

6. The above are not a thorough list for typos, the authors should carefully review the paper.

Reviewer #2: The authors applied Mendelian randomization (MR) approach to conduct a rapid and cost-effective study to examine the potential causal relationship between alcoholic liver disease (ALD) and bone mineral density (BMD) at four locations. Although the MR approach depends on assumptions that are not testable or provable, the authors attempted to perform sensitivity tests (including Egger, PRESSO, and RAPS) which estimate the association using different assumptions about the causal structure. Overall, the approach is largely appropriate.

My major concern is the unclear hypothesis of the study. The authors identified the research gap based on the limited knowledge on the risk factors for BMD and osteoporosis. They stated that liver function may affect the BMD status and gave one example that ALD has been described as an independent risk factor of reduced BMD. Given all these, they conducted MR analysis to test the relationship between ALD and BMD. However, the analysis did not fully address the research gap. The major chronic liver disease worldwide is nonalcoholic fatty disease (NAFLD). If liver function is a major contributor, the authors should also perform the MR analysis to test if NAFLD is associated with BMD. In addition, the cause of ALD is alcohol drinking, the authors need to test how alcohol drinking may affect BMD. Simply removing the alcohol-associated SNPs is not sufficient.

Other comments.

1. MR analysis provides somewhat robust conclusion regarding relationships; however, it cannot give a definite conclusion on causal association. Please remove “causal” from title.

2. There are a few editing issues, e.g., the abbreviation BMD in abstract is in a wrong place.

3. Please clarify what criteria were used to exclude “variants that were strongly correlated with the outcome to satisfy the hypothesis that instrumental variables were only correlated with the results by the exposure.”

4. In methods, the authors removed SNPs for confounders. The authors need to explain what factors/parameters/criteria have been used to evaluate socioeconomic status, and add SNPs related to this status to supplemental Tables.

5. Also in the “removing confounders” section, the authors stated that “Finally, null SNPs were excluded from the selected IVs for ALD…”. What are null SNPs?

6. In the Method “Two-sample MR analyses” section, authors stated that “Even if none of the IVs are correct, the MR-Egger technique nevertheless yields a more reliable estimate…. total estimate originates from incorrect genetic variations …”. These sentences are confusing. How come using incorrect SNPs can lead to a correct conclusion? Do the authors mean that these SNPs do not meet the MR assumptions?

7. The bi-directional design is interesting; however, the sample size of ALD is far larger than that for BMD (Supplemental Table 1). The authors should provide power calculation for all these analyses.

8. Resolution of the plots is low.

9. As what the authors have pointed out (in Discussion), validity of IVs may be problematic, e.g., the association p value threshold is 5e-6, which is below the conventional GWAS significance.

6. PLOS authors have the option to publish the peer review history of their article (what does this mean?). If published, this will include your full peer review and any attached files.

Reviewer #1: No

Reviewer #2: No

---

## [Author Response · Author response to Decision Letter 0]

20 Jun 2023

Point-to-point Response

Reviewer 1#

1.The authors found that genetically determined ALD was associated with a decreasing trend of FN-BMD using the IVW (β = -0.03, 95% CI: -0.05 to -0.01, P = 0.0077). Although the p-value is small, it is better to specify a threshold for multiple testing.

Answer: Thanks for your insightful comments. As four different sites of bone density were considered, a Bonferroni correction was applied to counteract the problem of multiple comparisons, and a P value of <0.0125 (P= 0.05/4 = 0.0125) was considered statistically significant in this study. We have used this threshold for testing in our revised manuscript.

2. If multiple testing threshold was applied, then only the IVW method gave significant results, all the sensitivity analyses (4 tests) showed no causal association. In this case, is it still reasonable to conclude ALD was causally associated with FN-BMD? How would you explain the different results between IVW and those sensitivity analyses?

Answer: Thanks for your comments. The IVW approach has the strictest assumptions about instrumental variables and the highest statistical efficiency, so reasonably the IVW results were generally used as the primary analysis, complemented by other methodologies. If the direction of the effect estimated by these sensitivity analyses is consistent with IVW, the results of IVW were generally regarded as valid. This strategy has been widely used in previously published study[1].To make it clear, we summarized the assumptions and features of each approach in the following Table 1. Please let us know if there is any question.

3.In supplemental table 2, rs75978489 has EAF 0. Is this a typo or a rare variant? All of the variants in this table have remarkably high R squared. How was the R square calculated?

Answer: Thank you for your comments. EAF 0 is a typo error and we have revised it accordingly. Here, R2 = 2 × EAF× (1 - EAF) × beta2, where EAF is the effect allele frequency and beta is the estimated genetic effect on exposure [2]. 

4.In results part, the author wrote “according to our selection criteria, 15 SNPs were maintained as IVs for ALD”, while figure A and B only showed 11 SNPs?

Answer: Thank you for your comments. These 15 SNPs were only strongly associated with ALD according to the screening threshold. They are needed to go through a series of filtering steps before they could be used for MR analysis. First, outlier pleiotropic SNPs were excluded. Next, SNPs that were strongly correlated with the outcome or that were not matched in the summary outcome data were removed. The remaining SNPs were used in the subsequent MR analysis (Figure 2; Table S12). Therefore, the numbers were not matched. 

5.Given the null association for other BMDs, power calculations should be included to assess for risk of type 2 error.

Answer: Thanks a lot for pointing out a very important issue. In our revised manuscript, power calculations have been added (Table S20). 

6.The authors state that there is evidence for a causal effect between ALD and FN-BMD. Could this be further evaluated –eg is there evidence that liver disease treatment (or liver transplant), or people who quit drinking associated with a rise of FN-BMD?

Answer: Thank you very much for your insightful suggestion. We have searched the relevant literature and found no GWAS available in people treated for liver disease or in people who have abstained from alcohol. Therefore, we are unable to use MR methodology to further explore whether treatment of liver disease or abstinence from alcohol is associated with an increase in FN-BMD. However, your suggestion is very valuable, and we will investigate it in future studies.

7.in the discussion the fifth point, the authors state they cannot estimate the extent of overlap in the study. Could this be evaluated by checking the cohorts that each database included?

Answer: Thanks very much for your insightful comments. We reviewed exposure data for ALD and NAFLD in the FinnGen cohort, potential mediators (risk factors) in the UK Biobank cohort, and outcome BMD in the GEFOS, and added detailed cohort information to Table S1, with no sample overlap identified in our study. Therefore, we have deleted this point in the revised manuscript. Please let us know if there is any other concern.

Minor:

1. I would tone down the claims for causality from the observational data throughout. Significant genetic associations are consistent with causality but one cannot definitively claim causality as the authors imply throughout.

Answer: Thanks for your kind suggestion. We have toned down the claims for causality by changing the expression to “genetically predicted effect of A on B”. We have revised the manuscript thoroughly. 

2. Figure 1: Romoving should be removing

Answer: We have corrected it accordingly.

3. Page 15: weighted model -> weighted mode

Answer: We have corrected it accordingly.

4. Results: 138 SNPs for TB-BMD, while in supplemental table 3, 139 SNPs

Answer: Thank you for pointing out this issue. We have made a mistake and have corrected it in the new manuscript.

5. In strengths and limitations: excessive period after “variations as IVs”

Answer: Thank you for pointing out this issue. We have corrected it accordingly.

6. The above are not a thorough list for typos, the authors should carefully review the paper.

Answer: We apologize for all these errors which might have brought you discomfort when reviewing the manuscript. We have carefully revised them and the manuscript was polished by a professional proofreading editor.

#Reviewer 2

1.My major concern is the unclear hypothesis of the study. The authors identified the research gap based on the limited knowledge on the risk factors for BMD and osteoporosis. They stated that liver function may affect the BMD status and gave one example that ALD has been described as an independent risk factor of reduced BMD. Given all these, they conducted MR analysis to test the relationship between ALD and BMD. However, the analysis did not fully address the research gap. The major chronic liver disease worldwide is nonalcoholic fatty disease (NAFLD). If liver function is a major contributor, the authors should also perform the MR analysis to test if NAFLD is associated with BMD. In addition, the cause of ALD is alcohol drinking, the authors need to test how alcohol drinking may affect BMD. Simply removing the alcohol-associated SNPs is not sufficient.

Answer: Thanks a lot for such insightful comments. First of all, we have revised our introduction section to make the research gap clearer. Overall, 40% of osteoporosis are secondary to chronic diseases, including chronic liver diseases. Patients with chronic liver disease may experience abnormal metabolic changes in bones, known as hepatic osteodystrophy commonly manifested as decreased BMD and osteoporosis. ALD and NAFLD are two major types of chronic liver diseases. Previous observational studies have demonstrated reduced BMD and increased susceptibility to intertrochanteric and hip fractures in patients with ALD. In contrast, the correlation between NAFLD and osteoporosis remains controversial with both positive and null correlations having ever been reported previously. However, previous studies are largely based on an observational cross-sectional design which is susceptible to confounding bias and reverse causality, thus there is still a lack of genetic evidence confirming their causal relationship. Given the dramatically increasing prevalence of ALD and NAFLD, particularly NAFLD, it is of great clinical significance to elucidate their causal relationship with BMD and identify potential mediators for osteoporosis prevention and management. 

Therefore, in our revised manuscript, we also explored the genetically predicted NAFLD on BMD, and a null effect was found. In addition, we also took common risk factors into account and investigated whether they mediated the association between ALD or NAFLD and BMD. In the current study, we focus on alcohol intake, smoking, and serum 25 Hydroxy vitamin D (25-OHD) levels, and used multivariate MR and two-step MR to explore whether and to what extent they mediated the effect of ALD on BMD. Our results demonstrated alcohol intake and smoking did not have significant effects on BMD, while serum 25-OHD level served as a significant mediator between ALD and reduced BMD. We have revised the manuscript accordingly. Again, we sincerely appreciated your comments which have helped us improved the manuscript greatly. Please do let us know if there is any question.

Other comments.

1.MR analysis provides somewhat robust conclusion regarding relationships; however, it cannot give a definite conclusion on causal association. Please remove “causal” from title.

Answer: Thanks for your kind suggestion. We have revised the title accordingly.

2.There are a few editing issues, e.g., the abbreviation BMD in abstract is in a wrong place.

Answer: Thank you for pointing out these issues. We have carefully revised them and the manuscript was polished by a professional proofreading editor.

3.Please clarify what criteria were used to exclude “variants that were strongly correlated with the outcome to satisfy the hypothesis that instrumental variables were only correlated with the results by the exposure.

Answer: We excluded variants that were strongly correlated with the outcome (Bonferroni-corrected significance threshold: P <0.05 /number of SNPs) to satisfy the hypothesis that instrumental variables were only correlated with the results by the exposure, as described in previously published studies[8, 9].

4 and 5. In methods, the authors removed SNPs for confounders. The authors need to explain what factors/parameters/criteria have been used to evaluate socioeconomic status and add SNPs related to this status to supplemental Tables. Also in the “removing confounders” section, the authors stated that “Finally, null SNPs were excluded from the selected IVs for ALD…”. What are null SNPs?

Answer: Thanks for your kind suggestions. We used the “PhenoScanner” database to identify and exclude genetic variants associated with confounders in our previous manuscript. But thanks to your comments, we realized the drawbacks of this strategy, namely because it does not necessarily distinguish between horizontal and vertical pleiotropy, as only the former would bias the study of MR. Meanwhile, the exact biological function of many genetic variants is unknown[3]. Therefore, we have optimized the filtering step of the instrumental variables in our study and used the "RadialMR" package to identify outlier pleiotropic SNPs, resulting in improved genetics and robustness of our instrumental variables[4]. This strategy has been adopted in previous studies including some published in high-impact journals (Jing Guo et al. nature neuroscience. 2022)[5] and (Shi Yao et al. nature human behaviour. 2022)[6] . In summary, we used “RadialMR” for instrumental variable quality control as an alternative to using the PhenoScanner database to find confounding SNPs. In addition, we further used multivariate MR and two-step MR to explore whether and to what extent common risk factors influence the association of ALD with FN-BMD.

6. In the Method “Two-sample MR analyses” section, authors stated that “Even if none of the IVs are correct, the MR-Egger technique nevertheless yields a more reliable estimate…. total estimate originates from incorrect genetic variations …”. These sentences are confusing. How come using incorrect SNPs can lead to a correct conclusion? Do the authors mean that these SNPs do not meet the MR assumptions?

Answer: Thanks for pointing out this issue and we apologize for make you confused on these sentences. Slope coefficients from MR-Egger regressions provide consistent estimates of causal effects in the presence of horizontal pleiotropy. We have revised the manuscript accordingly.

7. The bi-directional design is interesting; however, the sample size of ALD is far larger than that for BMD (Supplemental Table 1). The authors should provide power calculation for all these analyses.

Answer: Thanks a lot for pointing out a very important issue. In our revised manuscript, power calculations have been added (Table S20). We also addressed the limitation of insufficient power which may preclude us for making reliable inference. Therefore, further studies with larger sample size and larger statistical power are needed to confirm our findings. 

8.Resolution of the plots is low.

Answer: Thanks for your comments. We have revised the plots to make it clearer. 

9. As the authors have pointed out in the Discussion, the validity of IVs may be problematic, e.g., the association p-value threshold is 5e-6, which is below the conventional GWAS significance.

Answer: Thanks for your comments. Regarding the selection of genetic instruments, only one SNP was identified for association with ALD and two SNPs for association with NAFLD if a conventional P< 5 × 10-8 were applied. However, a minimum of 10 instrumental variables is required to conduct MR studies, and too few SNPs would make it difficult to match instrumental variables in the outcomes. For this reason, we chose SNPs with a less stringent significance threshold of 5 ×10–6, as described in earlier studies[10, 11]. However, the use of a less strict P value did not bring weak instrument bias; all the F statistics of these instrument variables were greater than 10 in the current study. We have revised the discussion part accordingly.

1. Cai J, Wei Z, Chen M, He L, Wang H, Li M, et al. Socioeconomic status, individual behaviors and risk for mental disorders: A Mendelian randomization study. Eur Psychiatry. 2022;65(1):e28. Epub 2022/04/19. doi: 10.1192/j.eurpsy.2022.18. PMID: 35431011.

2. Meddens SFW, de Vlaming R, Bowers P, Burik CAP, Linner RK, Lee C, et al. Genomic analysis of diet composition finds novel loci and associations with health and lifestyle. Mol Psychiatry. 2021;26(6):2056-69. Epub 2020/05/13. doi: 10.1038/s41380-020-0697-5. PMID: 32393786.

3. Wu F, Huang Y, Hu J, Shao Z. Mendelian randomization study of inflammatory bowel disease and bone mineral density. BMC Med. 2020;18(1):312. Epub 2020/11/11. doi: 10.1186/s12916-020-01778-5. PMID: 33167994.

4. Bowden J, Spiller W, Del Greco MF, Sheehan N, Thompson J, Minelli C, et al. Improving the visualization, interpretation and analysis of two-sample summary data Mendelian randomization via the Radial plot and Radial regression. Int J Epidemiol. 2018;47(4):1264-78. doi: 10.1093/ije/dyy101. PMID: 29961852.

5. Guo J, Yu K, Dong SS, Yao S, Rong Y, Wu H, et al. Mendelian randomization analyses support causal relationships between brain imaging-derived phenotypes and risk of psychiatric disorders. Nat Neurosci. 2022;25(11):1519-27. Epub 2022/10/11. doi: 10.1038/s41593-022-01174-7. PMID: 36216997.

6. Yao S, Zhang M, Dong SS, Wang JH, Zhang K, Guo J, et al. Bidirectional two-sample Mendelian randomization analysis identifies causal associations between relative carbohydrate intake and depression. Nat Hum Behav. 2022;6(11):1569-76. Epub 2022/07/20. doi: 10.1038/s41562-022-01412-9. PMID: 35851841.

7. Kurki MI, Karjalainen J, Palta P, Sipila TP, Kristiansson K, Donner KM, et al. FinnGen provides genetic insights from a well-phenotyped isolated population. Nature. 2023;613(7944):508-18. Epub 2023/01/19. doi: 10.1038/s41586-022-05473-8. PMID: 36653562.

8. Larsson SC, Burgess S, Michaelsson K. Association of Genetic Variants Related to Serum Calcium Levels With Coronary Artery Disease and Myocardial Infarction. JAMA. 2017;318(4):371-80. Epub 2017/07/26. doi: 10.1001/jama.2017.8981. PMID: 28742912.

9. Song X, Wang C, Wang T, Zhang S, Qin J. Obesity and risk of gestational diabetes mellitus: A two-sample Mendelian randomization study. Diabetes Res Clin Pract. 2023;197:110561. Epub 2023/02/05. doi: 10.1016/j.diabres.2023.110561. PMID: 36738839.

10. Zou XL, Wang S, Wang LY, Xiao LX, Yao TX, Zeng Y, et al. Childhood Obesity and Risk of Stroke: A Mendelian Randomisation Analysis. Front Genet. 2021;12:727475. Epub 2021/12/07. doi: 10.3389/fgene.2021.727475. PMID: 34868204.

11. Gao X, Meng LX, Ma KL, Liang J, Wang H, Gao Q, et al. The bidirectional causal relationships of insomnia with five major psychiatric disorders: A Mendelian randomization study. Eur Psychiatry. 2019;60:79-85. Epub 2019/06/25. doi: 10.1016/j.eurpsy.2019.05.004. PMID: 31234011.

---

## [Decision Letter · Decision Letter 1]

7 Aug 2023

PONE-D-23-08785R1The associations of alcoholic liver disease and nonalcoholic fatty liver disease with bone mineral density and the mediation of serum 25-Hydroxyvitamin D: a bidirectional and two-step Mendelian randomizationPLOS ONE

Dear Dr. Zhang,

Thank you for submitting your manuscript to PLOS ONE. After careful consideration, we feel that it has merit but does not fully meet PLOS ONE’s publication criteria as it currently stands. Therefore, we invite you to submit a revised version of the manuscript that addresses the points raised during the review process.

We look forward to receiving your revised manuscript.

Kind regards,

Chunyu Liu, PhD

Academic Editor

PLOS ONE

Journal Requirements:

Please address all of the comments by the reviewers

Reviewers' comments:

Reviewer's Responses to Questions

**Comments to the Author**

1. If the authors have adequately addressed your comments raised in a previous round of review and you feel that this manuscript is now acceptable for publication, you may indicate that here to bypass the “Comments to the Author” section, enter your conflict of interest statement in the “Confidential to Editor” section, and submit your "Accept" recommendation.

Reviewer #1: (No Response)

Reviewer #3: All comments have been addressed

2. Is the manuscript technically sound, and do the data support the conclusions?

Reviewer #1: Partly

Reviewer #3: Yes

3. Has the statistical analysis been performed appropriately and rigorously? 

Reviewer #1: Yes

Reviewer #3: Yes

4. Have the authors made all data underlying the findings in their manuscript fully available?

Reviewer #1: Yes

Reviewer #3: Yes

5. Is the manuscript presented in an intelligible fashion and written in standard English?

Reviewer #1: Yes

Reviewer #3: Yes

6. Review Comments to the Author

Reviewer #1: 1. Upon reviewing Tables S2 and S3, it becomes evident that some of the variants exhibit unusually high R2 values. As a result, it is imperative for the authors to carefully reassess their calculations and the formulas utilized in the analysis. Such remarkably elevated R2 values may warrant closer scrutiny to ensure the accuracy and reliability of the reported results.

2. Consistency in the number of significant figures throughout the entire manuscript is crucial. Therefore, it is essential to ensure that the chosen number of significant figures remains uniform across all sections and calculations presented in the paper.

For example IVW beta = -0.0288; 95% CI: -0.0488, -0.0087; P = 0.005, if adopting three significant figures, the values should be presented as IVW beta = -0.0288; 95% CI: -0.0488, -0.00870; P = 0.00500

The manuscript currently contains a mix of two, three, and four significant figures, leading to inconsistency in reporting. To rectify this, the authors have the flexibility to choose which format (two, three, or four significant figures) to adopt throughout the paper.

Reviewer #3: (No Response)

7. PLOS authors have the option to publish the peer review history of their article (what does this mean?). If published, this will include your full peer review and any attached files.

Reviewer #1: No

Reviewer #3: No

---

## [Author Response · Author response to Decision Letter 1]

9 Aug 2023

Dear Reviewers,

We would like to express our heartfelt thanks to you for the insightful comments. We have revised the manuscript according to your comments. Please find our point-to-point responses in the following. If there is any other concern, please don’t hesitate to let us know.

Point-to-point Response

Reviewer 1#

1.Upon reviewing Tables S2 and S3, it becomes evident that some of the variants exhibit unusually high R2 values. As a result, it is imperative for the authors to carefully reassess their calculations and the formulas utilized in the analysis. Such remarkably elevated R2 values may warrant closer scrutiny to ensure the accuracy and reliability of the reported results.

Answer: Thank you for your insightful comments. We carefully reviewed Supplementary Tables 2 and 3 and found that, as you said, two genetic variants, rs2294915 and rs738408, showed abnormally high R2 values. Therefore, we carefully checked the data of these two SNPs and found their EAF and beta values were incorrectly filled in Supplementary Tables 2 and 3, resulting in abnormally high R2 values. We have corrected them in the revised manuscript. Meanwhile, the values of other SNPs were also carefully checked to ensure the accuracy of our results. We apology for the careless mistake.

2.Consistency in the number of significant figures throughout the entire manuscript is crucial. Therefore, it is essential to ensure that the chosen number of significant figures remains uniform across all sections and calculations presented in the paper.

For example IVW beta = -0.0288; 95% CI: -0.0488, -0.0087; P = 0.005, if adopting three significant figures, the values should be presented as IVW beta = -0.0288; 95% CI: -0.0488, -0.00870; P = 0.00500. The manuscript currently contains a mix of two, three, and four significant figures, leading to inconsistency in reporting. To rectify this, the authors have the flexibility to choose which format (two, three, or four significant figures) to adopt throughout the paper.

Answer: Thanks a lot for your comments. We have revised our manuscript and used three significant figures throughout the manuscript.

Response to journal

Answer: Thank you for your comments. We have re-examined the references in our manuscript and did find that some of the references had incomplete information, as you mentioned. We have revised the manuscript accordingly. No references to the retracted paper were cited in our manuscript. In addition, the link to the second reference we cited was accidentally lost, so we have updated the corresponding reference and revised the sentence "affects approximately 200 million individuals worldwide" to "affects approximately 500 million men and women worldwide".

---

## [Decision Letter · Decision Letter 2]

2 Oct 2023

The associations of alcoholic liver disease and nonalcoholic fatty liver disease with bone mineral density and the mediation of serum 25-Hydroxyvitamin D: a bidirectional and two-step Mendelian randomization

PONE-D-23-08785R2

Dear Dr. Zhang,

We’re pleased to inform you that your manuscript has been judged scientifically suitable for publication and will be formally accepted for publication once it meets all outstanding technical requirements.

Kind regards,

Hiroshi Kaji

Academic Editor

PLOS ONE

Additional Editor Comments (optional):

Reviewers' comments:

Reviewer's Responses to Questions

**Comments to the Author**

1. If the authors have adequately addressed your comments raised in a previous round of review and you feel that this manuscript is now acceptable for publication, you may indicate that here to bypass the “Comments to the Author” section, enter your conflict of interest statement in the “Confidential to Editor” section, and submit your "Accept" recommendation.

Reviewer #1: All comments have been addressed

Reviewer #3: (No Response)

2. Is the manuscript technically sound, and do the data support the conclusions?

Reviewer #1: Yes

Reviewer #3: (No Response)

3. Has the statistical analysis been performed appropriately and rigorously? 

Reviewer #1: Yes

Reviewer #3: (No Response)

4. Have the authors made all data underlying the findings in their manuscript fully available?

Reviewer #1: Yes

Reviewer #3: (No Response)

5. Is the manuscript presented in an intelligible fashion and written in standard English?

Reviewer #1: Yes

Reviewer #3: (No Response)

6. Review Comments to the Author

Reviewer #1: Please check this paper "A Multivariate Genome-Wide Association Analysis of 10 LDL Subfractions, and Their Response to Statin Treatment, in 1868 Caucasians" S1 text for calculating proportion of variance explained.

Reviewer #3: (No Response)

7. PLOS authors have the option to publish the peer review history of their article (what does this mean?). If published, this will include your full peer review and any attached files.

Reviewer #1: No

Reviewer #3: No

---

## [Editor Report · Acceptance letter]

11 Oct 2023

PONE-D-23-08785R2 

The associations of alcoholic liver disease and nonalcoholic fatty liver disease with bone mineral density and the mediation of serum 25-Hydroxyvitamin D: a bidirectional and two-step Mendelian randomization 

Dear Dr. Zhang:

I'm pleased to inform you that your manuscript has been deemed suitable for publication in PLOS ONE. Congratulations! Your manuscript is now with our production department. 

Kind regards, 

on behalf of

Dr. Hiroshi Kaji 

Academic Editor

PLOS ONE